# Kaempferol Alleviates Mitochondrial Damage by Reducing Mitochondrial Reactive Oxygen Species Production in Lipopolysaccharide-Induced Prostate Organoids

**DOI:** 10.3390/foods12203836

**Published:** 2023-10-20

**Authors:** Myeong Joon Lee, Yeonoh Cho, Yujin Hwang, Youngheun Jo, Yeon-Gu Kim, Seung Hwan Lee, Jong Hun Lee

**Affiliations:** 1Department of Food Science and Biotechnology, Gachon University, Seongnam 13120, Republic of Korea; lmg961113@gachon.ac.kr (M.J.L.); 970907cyo@gachon.ac.kr (Y.C.); rnddl0323@gachon.ac.kr (Y.H.); 2Department of Urology, Yonsei University College of Medicine, Seoul 03722, Republic of Korea; hihi1541@yuhs.ac; 3Biotherapeutics Translational Research Center, Korea Research Institute of Bioscience and Biotechnology (KRIBB), 125 Gwahak-ro, Yuseong-gu, Daejeon 34141, Republic of Korea; dusrn9@naver.com; 4Department of Bioprocess Engineering, KRIBB School of Biotechnology, Korea University of Science and Technology (UST), 217 Gajeong-ro, Yuseong-gu, Daejeon 34141, Republic of Korea

**Keywords:** organoid, kaempferol, anti-inflammation, antioxidant, ROS, mitophagy, mitochondrial homeostasis

## Abstract

Common prostate diseases such as prostatitis and benign prostatic hyperplasia (BPH) have a high incidence at any age. Cellular stresses, such as reactive oxygen species (ROS) and chronic inflammation, are implicated in prostate enlargement and cancer progression and development. Kaempferol is a flavonoid found in abundance in various plants, including broccoli and spinach, and has been reported to exhibit positive biological activities, such as antioxidant and anti-inflammatory properties. In the present study, we introduced prostate organoids to investigate the protective effects of kaempferol against various cellular stresses. The levels of COX-2, iNOS, p-IκB, a pro-inflammatory cytokine, and ROS were increased by LPS treatment but reversed by kaempferol treatment. Kaempferol activated the nuclear factor erythroid 2-related factor 2(Nrf2)-related pathway and enhanced the mitochondrial quality control proteins PGC-1α, PINK1, Parkin, and Beclin. The increase in mitochondrial ROS and oxygen consumption induced by LPS was stabilized by kaempferol treatment. First, our study used prostate organoids as a novel evaluation platform. Secondly, it was demonstrated that kaempferol could alleviate the mitochondrial damage in LPS-induced induced prostate organoids by reducing the production of mitochondrial ROS.

## 1. Introduction

Prostatitis and benign prostatic hyperplasia (BPH) are the most common benign prostate diseases. Prostatitis has a high incidence in men regardless of age [1], and its causes include infection with pathogens via the lower urinary tract and urinary reflux [2,3]. It is an inflammatory disease that is characterized by severe pain and dysuria. Despite being a common disease, it is not easy to treat, which seriously reduces the quality of life.

The incidence of BPH increases with age, and androgens are also regarded as the cause of BPH [4]. It is usually characterized by lower urinary tract symptoms (LUTS), an enlarged prostate, and decreased urine output. It has been shown that oxidative stress caused by prostate disease can stimulate stromal and epithelial cells to hyperproliferate prostate tissue cells [5]. It has also been suggested that histological inflammation is directly or indirectly involved in the initiation and progression of prostatic hypertrophy and prostate cancer (PCa) by inducing rapid upregulation of reactive oxygen species (ROS) and repeated damage repair cycles of prostate tissue through the secretion of cytokines [5,6]. Increased oxidative stress may alter the normal functioning of the prostate, producing more ROS and disrupting the cellular antioxidant system, leading to mitochondrial dysfunction and mutations [7]. Thus, we hypothesized that relieving oxidative stress might modulate prostatic disease prevention and protective activity.

For prostate organoids, the epithelium consists of a basal layer (outline) that specifically expresses basal prostate markers such as p63, cytokeratin 14 (CK14), cytokeratin 5 (CK5), androgen receptor (AR), cytokeratin 18 (CK18), and cytokeratin 8 (CK8), which form a cystic structure consisting of a luminal (inner line) layer [8]. Prostate organoids can maintain a prostate-like structure in vitro but do not express neuroendocrine cells [9]. R-spondin and Wnt are known to be involved in prostate development, and the Wnt signaling pathway is essential for prostate organoid development [10]. The R-spondin receptors Lgr4 and Lgr5 are essential for prostate development and differentiation, and organoids express these two factors [11,12]. Current prostate research is focused on targeting prostate cancer and is limited by the lack of suitable in vitro model systems. Therefore, organoid culture systems have emerged as alternative platforms.

Kaempferol (3,4′,5,7-tetrahydroxyflavone), a flavonoid found in various plants, such as broccoli and spinach, has been reported to have several positive pharmacological activities, including potent anti-inflammatory and antioxidant abilities [13,14,15]. Kaempferol has four phenolic hydroxyl groups, of which 3′-hydroxyl group has been reported to have excellent activity in suppressing the oxidation of free radicals (Figure 1) [16]. In addition, transcription factors via the nuclear erythrocyte 2-associated factor 2(Nrf2)-antioxidant response element (ARE) signaling pathway are activated against ROS-induced oxidative stress in various cells, demonstrating antioxidant activity and cytoprotective effects [17,18]. Mitochondria are important organelles for maintaining cell homeostasis, and mitochondrial biosynthesis and mitophagy are quality-control processes that regulate mitochondrial content and metabolic homeostasis. Since mitochondria are the cellular organelles where oxidative phosphorylation directly takes place, mitochondrial DNA is susceptible to damage and these damaged mitochondria undergo the mitophagy process. It is known that an imbalance between mitochondrial proliferation and degradation in quality control including mitophagy dysfunction can increase the amount of damaged mitochondria in cells, reduce metabolic efficiency, and accumulate ROS, leading to cell death [19,20,21]. Kaempferol activates NAD-dependent deacetylase sirtuin-1 (SIRT1) in vitro to induce peroxisome proliferator-activated receptor gamma coactivator 1-alpha (PGC-1α) and PTEN-induced kinase 1 (PINK1) proteins involved in mitochondrial homeostasis [22]. However, to the best of our knowledge, there are no previous studies related to the crosstalk between Nrf2 and mitochondrial regulatory proteins in prostate cancer. Therefore, we wanted to determine the effects of kaempferol on mitochondrial metabolism and antioxidant activity in LPS-induced prostate organoids.

Specifically, the current study aimed to confirm the anti-inflammatory effect and ROS inhibition capacity of kaempferol on LPS-induced stress in prostate organoids. We also hypothesized and demonstrated that these protective and inhibitory effects are mediated by the Nrf2-mediated antioxidant response to kaempferol and the maintenance of mitochondrial homeostasis via the improvement of mitochondrial function.

## 2. Materials and Methods

### 2.1. Chemical Reagents

Kaempferol, dihydrotestosterone (DHT), and N-acetyl-1-cysteine (NAC) were purchased from Sigma-Aldrich (St. Louis, MO, USA). 2′,7′-dichlorodihydrofluorescein diacetate (DCF-DA), Mitotracker, and MitoSOX were purchased from Invitrogen™ (Massachusetts, CA, USA). Human epidermal growth factor (EGF) was purchased from Proteintech (Rosemont, IL, USA). A83-01 and Y-27632 dihydrochloride were purchased from Tocris Biosciences (Bristol, UK). Collagenase type II was purchased from Gibco (Waltham, MA, USA). 4′,6-diamidino-2-phenylindole dihydrochloride (DAPI) was purchased from VectorLab (Newark, CA, USA). CK8, E-cadherin, NOS2, COX-2, phos-pho-IκBα, total IκBα, Nrf2, HO-1, NQO-1, Beclin, PINK1, Parkin, PGC1α, and β-actin were purchased from Santa Cruz Biotechnology (Dallas, TX, USA). Antibodies against CK5 and Lgr5 were purchased from Abcam Inc. (Cambridge, UK).

### 2.2. Organoid Formation

The mice used in the experiment were purchased from Orient Bio (Orient Bio, Gyeonggi-do, Republic of Korea). After euthanizing 7-week-old male mice with CO_2_, the prostate was separated to remove unnecessary connective tissue and blood vessels. The isolated prostate tissue was cut into small pieces (1 mm^3^) and washed with washing medium (DMEM + P/S, 1% FBS). Digestion was then performed for 1 h in a shaking water bath with basal media (advanced DMEM + P/S, GlutaMAX, HEPES) containing 5 mg/mL collagenase type II and 10 μM Y-27632 dihydrochloride, and the primary cells were harvested via filtering using a 70 μM cell strainer (Corning, NY, USA). After sufficient washing, the harvested cells were plated in a dome-shape on a 24-well plate (SPL, Gyeonggi-do, Republic of Korea) with culture medium (with Y-27632 dihydrochloride) and Matrigel. When the Matrigel solidified, 600 μL of culture medium (with Y-27632 dihydrochloride) containing various growth factors was added to each well. The culture medium was changed every 2–3 days, and its composition is provided in Table 1. Cultured organoids were passaged at 70–80% once a week using TrypLE express (Gibco, Waltham, MA, USA) treatment. Media containing Y-27632 dihydrochloride should be used after plating and for up to 5–7 days after passage, and culture media without Y-27632 dihydrochloride should be used after the set period. Representative images of prostate organoids were acquired on days 0, 2, 4, and 8 after seeding, using an inverted microscope (Olympus CKX53; Olympus Corporation, Japan).

### 2.3. LPS Treatment in Organoids

After subculturing, prostate organoids were seeded in a 24-well culture plate and maintained at 37 °C and 5% CO_2_ until the organoids grew (2–3 days after seeding). Organoids were then pre-treated with kaempferol for 24 h, and then 10 μg/mL LPS (L2880, Sigma-Aldrich, MO, USA) and 20, 40, and 80 μM kaempferol were co-treated in each well for 24 h. Organoids not treated with LPS were used as controls for comparative analysis, and negative controls were treated with LPS only. In the text, the kaempferol-treated group is abbreviated as KF. After co-culture, organoids were harvested after washing with cold PBS.

### 2.4. Immunofluorescence

The media was removed from the organoid culture and gently washed with cold PBS to harvest the organoids. After fixing with 4% formaldehyde for 30 min at 25 °C, the cells were washed three times with cold PBS. After permeabilization at 25 °C with blocking buffer (3% BSA + 0.5% Triton X-100 in 1× PBS) for 1 h, primary antibodies, rabbit anti-CK5, rabbit anti-Lgr5, mouse anti-CK8, and mouse anti-E-cadherin were diluted 1:200. Overnight incubation of organoids with the primary antibody at 4 °C. Each well was washed thrice with PBS. Secondary antibodies, goat anti-rabbit IgG Alexa Fluor 488 (Abcam, UK) and goat anti-mouse IgG Alexa Fluor 568 (Invitrogen, MA, USA), were diluted at a ratio of 1:1000 and incubated at 25 °C for 1 h. After washing three times with PBS, the cells were stained with DAPI with the mounting solution and analyzed with a fluorescence microscope (Olympus BX53; Olympus Corporation, Tokyo, Japan).

### 2.5. Western Blot Analysis

The media was removed from the organoid culture and gently washed with cold PBS to harvest the organoids. The organoid pellet in the tube was dissolved on ice for 30 min using Nonidet P-40 (NP40, Invitrogen, MA, USA) and a protease inhibitor cocktail (Gendepot, TX, USA). The cell lysates were centrifuged at 16,000 rpm for 9 min at 4 °C. The supernatant was then transferred to a new microcentrifuge tube. Protein content was quantified using bicinchoninic acid (BCA) assay using a Pierce BCA Protein Assay Kit (Thermo Scientific, MA, USA). Equal concentrations of protein (20 μg) were separated by 12% sodium dodecyl sulfate–polyacrylamide gel electrophoresis (SDS-PAGE) and transferred to a polyvinylidene fluoride (PVDF) membrane (Merck Millipore, Germany). The membrane was blocked with a blocking buffer (3% bovine serum albumin in Tris-buffered saline Tween-20 (TBST) 0.5%) for 1 h at room temperature. The membrane was maintained overnight at 4 °C with primary antibodies NOS2 (1:1000), COX-2 (1:1000), p-IκBα (1:1000), IκBα (1:1000), Nrf2 (1:1000), HO-1 (1:1000), NQO-1 (1:1000), Beclin (1:1000), PINK1 (1:1000), Parkin (1:1000), PGC1α (1:1000), and β-actin (1:1000) and then washed three times with TBST. The membrane was then incubated with HRP-conjugated anti-IgG antibody (Promega, WI, USA). Then, the ECL solution (Dynebio, Republic of Korea) and Amersham imager 600 (GE Healthcare, IL, USA) were used to visualize the percentage of protein quantified using the ImageJ program.

### 2.6. Total RNA Extraction and RT-qPCR

Total RNA was isolated from the prostate organoids using RNeasy Mini Kits (QIAGEN, Hilden, Germany) according to the manufacturer’s protocol. RLT buffer (350 µL) (QIAGEN, Germany) and 70% ethanol were added to the organoid pellet in a microcentrifuge and the pellet was dissolved via pipetting. The lysate was collected in a 2 mL collection tube and centrifuged at 4 °C at 8000× *g* for 15 s. The supernatant was removed, 700 µL of buffer RW1 (QIAGEN, Germany) was added, and then it was centrifuged at 8000× *g* for 15 s. The supernatant was removed, washed twice with buffer RPE (QIAGEN, Germany), and replaced with a 1.5 mL collection tube. RNase-free water (20 µL) was added to the pellet and allowed to stand for 2 min until RNA dissolved. The supernatant was transferred to a new tube and centrifuged at 8000× *g* for 1 min. Purity and quantification of total RNA were performed using NanoDrop 2000 (Thermo Scientific, MA, USA). The extracted total RNA was synthesized into cDNA using a combination of dNTPs (Enzynomics, Daejeon, Republic of Korea), RNase inhibitor (Enzynomics, Republic of Korea), and M-MLV RT 5X Buffer (Promega, Madison, WI, USA) using reverse transcriptase (Promega, WI, USA). The expression of TNF-α, IL-6, IL-1β, COX-2, Nrf2, HO-1, NQO1, SOD1, and GAPDH was determined via RT-qPCR using TOPreal™ qPCR 2X Premix. All qPCR analyses were performed using QuantStudio 3 Real-Time PCR instrument (Thermo Scientific, MA, USA). The results were based on GAPDH expression, and the expression levels were measured using the ΔΔCt method. The primer sequences are listed in Table 2. TNF-α, IL-6, IL-1β, and COX2 primers (57 °C) and Nrf2, HO-1, NQO-1, and SOD1 primers (54 °C) were repeated 40 times.

### 2.7. Measurement of Intracellular ROS Production Using DCF-DA

Intracellular ROS generation was measured using the DCF-DA fluorescent dye. After the dye was integrated into the cell, 2′,7′-dichlorodihydrofluorescein diacetate (H2DCF-DA, Invitrogen, MA, USA) was rapidly oxidized to fluorescent 2′,7′-dichlorofluorescein (DCF) in the presence of intracellular ROS. Prostate organoids were cultured for 2–3 days after seeding in 96-well plates (SPL, Gyeonggi-do, Republic of Korea). When organoids were sufficiently grown, they were treated using various concentrations of kaempferol for 24 h. Kaempferol and LPS were co-treated at 37 °C and 5% CO_2_ for 23 h and then incubated with 10 μM H_2_DCF-DA in aluminum foil for 1 h. After removing the medium, the organoids were washed twice with cold PBS to remove as much Matrigel as possible, washed once with pre-warmed PBS, and analyzed using a plate reader (ex: 484 mm/em:535 mm). In an independent experiment, organoid pellets treated with KF + LPS were harvested using cold PBS and incubated using 10 μM of H_2_DCF-DA in an aluminum foil for 1 h. The pellet, from which the supernatant was removed, was incubated using DAPI for 30 min at 25 °C, and then the fluorescence was examined using a fluorescence microscope by covering it with glass.

### 2.8. Detection of Mitochondria and Mitochondrial ROS

Mitochondria are a major source of intracellular ROS, mainly produced in the electron transport chain, and include peroxides and superoxides. MitoTracker is a dye that specifically stains the mitochondria of living cells, whereas MitoSOX is a fluorescent dye that labels mitochondrial superoxide in living cells. It penetrates the cell, is oxidized by mitochondrial peroxide, and binds to nucleic acids, which can be visualized and measured. Prostate organoids were seeded in 24-well plates and cultured for 2–3 days. When organoids were sufficiently grown, they were treated with kaempferol 40 μM for 24 h. Next, the medium was removed, and kaempferol and LPS were co-treated at 37 °C and 5% CO_2_ for 24 h, harvested with cold PBS, and washed twice. To accumulate the fluorescent probe in the mitochondria, the cells were resuspended in PBS containing 100 nM Mitotracker Green (M7514, Invitrogen) and 1 μM MitoSOX (M36008, Invitrogen), covered with aluminum foil, and incubated for 30 min. After incubation, the supernatant was removed, and the organoids were attached to a coverslip and visualized using Eclipse C1si (Nikon, Japan).

### 2.9. Measurement of Oxygen Consumption Rate through Mitochondrial Metabolism

Mitochondrial respiration was assessed using a Seahorse XFp analyzer (Agilent Technologies, Santa Clara, CA, USA). Before analysis, the Seahorse XFp analyzer was turned on and warmed for one day. The Seahorse XF Calibrant flux cartridge (Agilent Technologies, CA, USA) with D.W was placed in a non-CO_2_ incubator at 37 °C overnight. Prostate organoids were cultured in Seahorse XFp cell culture miniplates (Agilent Technologies, CA, USA) using a complete medium for 2–3 days and then pretreated with kaempferol (40 μM) at 37 °C and 5% CO_2_ for 24 h. After the medium was removed, kaempferol and LPS were co-cultured for 24 h. D-glucose, L-glutamine, and sodium pyruvate (4.5 g/L each) were supplemented with Seahorse XFp DMEM (Agilent Technologies, CA, USA) at pH 7.4 to make an assay medium. After replacing the existing medium with fresh medium, incubated for 30–60 min in a non-CO_2_ incubator at 37 °C. Oligomycin (1.5 µM), trifluoromethoxy carbonyl cyanide 4-phenylhydrazone (FCCP) (0.5 µM), and rote-none/antimycin A (0.5 µM) were injected into the flux cartridge injection port and placed on the analyzer tray to activate calibration. After calibration, cell culture plates were loaded. Specific mitochondrial inhibitors and uncouplers were sequentially injected, according to the manufacturer’s instructions. All measurements were normalized to the total protein concentration using the Pierce BCA Protein Assay Kit.

### 2.10. Statistical Analysis

All data are expressed as mean ± standard deviation (SD). Statistical significance was determined using Student’s *t*-test, and the results were considered statistically significant at *p* < 0.05. The experiments were performed at least three times.

## 3. Results

### 3.1. Establishment of Mouse Prostate Organoid Expressing Basal and Luminal Epithelial Layers

Organoids were established from mouse prostate for effective prostate studies. For prostate organoid culture, dissociated cells were embedded in Matrigel with an added growth medium, and as the culture progressed, the organoids increased in size and formed a circular structure (Figure 2A). 

The epithelial-specific markers, cytokeratin5 (CK5), and cytokeratin8 (CK8) were identified using immuno-fluorescence staining. The expression of CK5 and CK8 was observed via immunofluorescence staining as shown in Figure 2B, suggesting phenotypic similarity between the constructed organoids and target organs. The expression of Lgr5 (green) and epithelial E-cadherin (red) in organoids was observed with an Olympus BX53 fluorescence microscope. It was confirmed that the prostate organoids expressed Lgr5 and the epithelial marker expressed E-cadherin (Figure 2C).

### 3.2. Kaempferol Inhibits iNOS, COX-2, p-IκB, and Inflammatory Cytokines in LPS-Stimulated Prostate Organoids

To examine the anti-inflammatory effect of kaempferol in prostate organoids, the inflammation of prostate organoids was induced by LPS, and the expression of inflammatory factors such as iNOS, COX-2, and p-IκB regulated by kaempferol was observed. As shown in Figure 3, the protein expression of iNOS, COX-2, and p-IκB was increased in the LPS-treated group (Figure 3A–D). The kaempferol-treated group was incubated alone for 24 h and then co-cultured with LPS. The expression of iNOS and COX-2, which increased during LPS treatment, decreased with kaempferol concentration (Figure 3B,C). In addition, the p-IκB/IκB ratio was significantly decreased (Figure 3D). 

Additionally, to investigate the inflammatory markers regulated by kaempferol at the gene level of prostate organoids, the mRNA expression of pro-inflammatory cytokines, such as TNF-α, IL-6, IL-1β, and COX-2, was observed (Figure 3E–H). Kaempferol reduced cytokine expression in a concentration-dependent manner, which was identical to the protein level (Figure 3E–G). However, the expression of COX-2 was the highest at 20 μM kaempferol, which was considered to have no significant effect on COX-2 mRNA suppression at a low concentration of kaempferol (Figure 3H).

### 3.3. Kaempferol Is Effective in Inhibiting Intracellular ROS Produced by LPS

Previously, kaempferol was shown to reduce inflammation-related factors, and it was expected that it could also reduce ROS levels based on its potent antioxidant activity. Therefore, we investigated intracellular ROS generation in organoids treated with various concentrations of kaempferol, using H_2_DCFDA, an oxidant-sensitive fluorescent probe (Figure 4). As shown in Figure 4, the LPS-induced intracellular ROS significantly increased by 38% compared to the control group (Figure 4A). A total of 20, 40, and 80 μM of kaempferol were treated with organoids that induced ROS using LPS, and ROS reduced by 40, 35, and 41%, respectively (Figure 4B). This suggests that kaempferol exerts a protective effect against LPS-induced oxidative stress.

### 3.4. Kaempferol Induces an Antioxidant Response by Activating the Nrf2 Pathway

As shown in Figure 4, kaempferol reduced inflammatory stress and effectively removed ROS. To determine whether it was due to antioxidant activity, the nrf2 pathway, a representative antioxidant pathway, was identified. We explored the expression levels of proteins to determine whether LPS and kaempferol treatment induced Nrf2-related antioxidant responses in organoids (Figure 5). Nrf2 decreased after LPS treatment and increased again in the kaempferol-treated group (Figure 5B). HO-1 and NQO-1 levels increased in the LPS-treated group and further increased in the kaempferol-treated group (Figure 5D,E). 

At the mRNA level, Nrf2, HO-1, NQO-1, and superoxide dismutase 1 (SOD1) were identified (Figure 5E–H). HO-1 increased upon LPS treatment (Figure 5F), and the expression of most factors above kaempferol (40 μM) was upregulated during co-treatment (Figure 5E,F,H). Therefore, kaempferol counteracted LPS-induced stress by activating Nrf2 and its downstream enzymes such as HO-1 and NQO-1.

### 3.5. Kaempferol Can Reduce the Generation of Mitochondrial ROS (mtROS)

Since it was shown that intracellular ROS production was inhibited and Nrf2-related antioxidant response was elevated by kaempferol previously (Figure 4 and Figure 5), we speculated that kaempferol would reduce LPS-stimulated mitochondria-generated ROS via antioxidant pathways. MitoTracker, a mitochondria-specific stain capable of labeling mitochondria, and MitoSox, which can detect live mitochondrial superoxide, were used to observe mitochondrial ROS generation. Thus, the generation of mtROS can be observed by contrasting the mitochondrial superoxide stained in red with the mitochondria stained in green. As shown in Figure 6, no red staining occurred in the control group but was reversed in the LPS-stimulated group. The generations of mtROS in the kaempferol-treated group were reduced, confirming that co-localization was reduced (Figure 6). Therefore, considering the results of intracellular ROS and mitochondrial peroxide, it can be inferred that kaempferol is effective in preventing and suppressing stress caused by LPS via the mechanism of reducing intracellular ROS, especially mtROS.

### 3.6. Reduced mtROS in Prostate Organoids Can Be Due to Enhanced Autophagy and an Increase in Mitochondrial Function by Kaempferol

To examine the effect of kaempferol on mitophagy and mitochondrial biosynthesis in prostate organoids, the mitochondrial biosynthesis biomarker, PGC1α, mitophagy markers such as PINK1 and Parkin, and the autophagy marker Beclin were also identified in prostate organoids (Figure 7). The factor involved in mitochondrial autophagy, Beclin, was sequentially upregulated in all treatment groups, and kaempferol concentration was most significant at 80 μM (Figure 7B). PINK1 and Parkin were slightly decreased upon LPS stimulation, which was increased by kaempferol treatment, and similarly increased significantly at 80 μM of kaempferol (Figure 7C,D). As shown in Figure 7E, an imbalance in the quality control process can be inferred from the increase in PGC1α levels during LPS treatment, and biosynthesis also increased during kaempferol treatment. Therefore, based on these results, it can be inferred that LPS induces the accumulation of damaged mitochondria in prostate organoids, and that kaempferol has a protective effect against dysfunction by upregulating disproportionate mitochondrial function.

### 3.7. ROS Increases Prostate Organoid Oxygen Consumption Rate (OCR), and Co-Treatment of Kaempferol Restores Mitochondrial Dysfunction via Mitophagy

It has been shown that the treatment of kaempferol effectively suppressed LPS-induced stress and improved mitochondrial function. In this context, we expected that the OCR increased by LPS could be decreased via the protective effects of kaempferol. Therefore, we measured the OCR related to mitochondrial function using a Seahorse bioenergetic metabolism analyzer after LPS and kaempferol treatment (Figure 8). In the LPS-induced group, the basal OCR was increased by 28% compared to maximal OCR, which was observed after FCCP injection. The co-treatment of kaempferol and LPS reduced the OCR to control levels (Figure 8A). In the case of OCR, cellular respiration, ATP, and cell viability are important factors. There have been studies showing that kaempferol is cytotoxic when treated at higher concentrations of 50 μm or more in various cells [23,24], and the above results show that it has significant activity even at concentrations below 50 μm. Therefore, we decided that 40 μm of kaempferol was the appropriate concentration. Interestingly, since LPS treatment lowered the maximum respiratory capacity, proton leak was higher in the LPS-treated group. This could be considered as a sign of LPS-induced mitochondrial damage (Figure 8B). LPS treatment increased the OCR in prostate organoids; therefore, increased OCR raised ROS generation in ETC. The co-treatment of kaempferol reduced OCR and the subsequent reduction in ROS could be expected.

## 4. Discussion

In the present study, first, we intended to verify prostate organoids as a test platform that overcomes the limitations of traditional 2D cell culture. Second, we aimed to observe improvements in mitochondrial function and antioxidant effects of kaempferol in LPS-induced prostate organoids.

Prostate organoid, a test platform that can observe biochemical changes occurring in real tissues using 3D mini-organs that go beyond the limits of 2D cell culture, was fabricated. Generally, the prostate epithelium consists of various cytokeratins, such as a basal (outside) layer expressing CK5 and a luminal (inside) layer expressing CK8 [8]. The Wnt signaling pathway is essential for prostate organoid development, and R-spondin protein expression activates the Wnt pathway, indicating a key role in signaling. Lgr5 and its homologs, Lgr4 and Lgr6, are biological receptors for R-spondin proteins and are expressed in the prostate [25]. As shown in Figure 2, the appropriate expression distribution of CK5 and CK8 in our fabricated organoids exhibited proper basal and luminal cell populations in organoids. In addition, the expression of Lgr5 indicated that organoids maintained stemness and differentiation. Thus, the organoids we fabricated have heterogeneity, composed of various types of cells constituting the tissue, and have characteristic elements mimicking the prostate.

LPS is known to activate an immune response via TLR4, leading to the release of proinflammatory cytokines and ROS [15]. LPS increased the protein expression of inflammatory protein factors, such as iNOS, COX-2, and p-IκB, and increased the mRNA expression of pro-inflammatory cytokines, such as TNF-α, IL-6, IL-1β, and COX-2. These over-expressed pro-inflammatory cytokines were significantly reduced in the kaempferol co-treated group, suggesting that kaempferol has an anti-inflammatory effect. These results showed a similar trend to the previous in vitro and in vivo experimental results treated with kaempferol, suggesting that the organoid model can be applied to the inflammation activity test for novel compounds [26,27].

Intracellular ROS can be generated from the outside by inflammatory cytokines produced to eliminate pathogens including bacteria, and can also be generated via mitochondrial oxidative phosphorylation [28,29]. These intracellular ROS can be balanced in our body by expressing antioxidant proteins such as GPx and SOD [30,31]; however, large amounts of ROS are not eliminated, causing various stresses such as lipid oxidation, DNA damage, and apoptosis [32]. As inflammatory cytokines and ROS are closely linked, we confirmed ROS using DCF-DA expression analysis. Fluorescence expression increased in organoids upon LPS treatment, and the fluorescence intensity was significantly decreased by kaempferol. To confirm that this ROS removal effect was mediated by antioxidant activity, Nrf2-related transcription factors were identified. The protein expression level of Nrf2 increased upon treatment with kaempferol, and the expression of HO-1 and NQO-1 also significantly increased. In addition, the mRNA expression levels of Nrf2, HO-1, NQO-1, and SOD1 were significantly upregulated upon kaempferol treatment. Kaempferol is known to lower oxidative stress and effectively scavenge ROS due to its potent antioxidant properties [25]. The antioxidant capacity of kaempferol has been shown to express the antioxidant system by upregulating SIRT1 to stimulate Nrf2 activation and target subfactors [33].

Nrf2 can regulate the expression of proteins involved in mitochondrial quality control as well as antioxidant properties. Mitochondria are important organelles for cellular energy metabolism, and the processes that regulate mitochondrial metabolic homeostasis include mitochondrial biosynthesis and mitophagy. The imbalances between mitochondrial biogenesis and mitophage processes lead to the development of various diseases, and these interactions should be closely maintained [34]. 

In general, excess free radicals produced during mitochondrial respiration are associated with inflammation and affect cell growth. In addition, an increase in ROS due to an inflammatory response causes oxidative damage to mitochondria and stimulates mitochondrial DNA (mtDNA), leading to increased mitophagy and apoptosis [35]. Damaged mitochondria cause an inflammatory response and generate additional ROS, which leads to continuous poor circulation. Mitochondrial removal through mitophagy and balancing ROS appear essential to maintain mitochondrial quality [36,37]. Whether kaempferol effectively inhibited ROS generation in mitochondria by increasing mitophagy was determined using MitoSox and MitoTracker. Mitochondrial ROS increased upon LPS treatment, and colocalization decreased, which is consistent with the intracellular ROS results. 

Mitochondrial biosynthesis induces mitochondrial mass and proliferation by expressing PGC1α, leading to efficient energy metabolism [38]. Mitophagy occurs in response to mitochondrial damage and normally leads to a pathway where PINK1 recruits Parkin to clear the mitochondria [39]. Kaempferol upregulates PGC1α via the activation of Nrf2, leading to mitochondrial biosynthesis and increasing PINK1 and Parkin to regulate mitochondrial quality control [30,40]. Furthermore, the upregulation of these two opposing actions of Nrf2 for mitochondrial homeostasis allows for quality control by regulating mitochondrial content in response to cellular stress and eliminating damaged mitochondria [21]. To confirm the regulation of mitochondrial biosynthesis and mitophagy by kaempferol, Beclin, PINK1, Parkin, and PGC1α were identified at the protein level. Beclin increased in both the LPS-treated group and the kaempferol co-treated group; PINK1 and Parkin decreased slightly in the LPS-treated group and were upregulated in the kaempferol co-treated group. In addition, PGC1α was slightly increased during LPS treatment and was significantly increased in the kaempferol co-treated group. LPS decreased mitophagy and increased the expression of PGC1α, which may indicate the accumulation of damaged mitochondria. This may indicate unstable mitochondrial homeostasis. Because kaempferol treatment stabilizes mitochondrial function, it is suggested that it improves mitochondrial function via mitophagy to maintain mitochondrial quality in prostate organoids (Figure 7). 

Mitochondria generate ROS via oxidative phosphorylation (OXPHOS) and the expression of pro-inflammatory cytokines via LPS stimulation triggers mitochondrial ROS generation [41]. Similarly, in RAW 264.7, LPS treatment increased the oxygen consumption rate, and in hyperplastic prostatic cells, kaempferol treatment decreased the oxygen consumption rate [42,43]. 

To measure the effects of LPS and kaempferol on mitochondrial respiratory metabolism, the oxygen consumption rate was measured using a metabolic analyzer. Oligomycin is used in the assay to inhibit ATP synthase (complex V) of the electron transport chain (ETC), which can determine the basic cellular energy demand. Carbonyl cyanide-4 (trifluoromethoxy) phenylhydrazone (FCCP) disrupts the proton gradient in the mitochondrial inner membrane. It can lead to a maximal oxygen consumption rate and measure reserve respiratory capacity, which may indicate the ability of the cell to respond to increased energy demands or stress. A mixture of rotenone and antimycin A blocks mitochondrial respiration by inhibiting complex I and complex III [44]. As shown in Figure 8, the LPS-treated group had higher basal respiration compared to its maximal respiration. Since the proton gradient collapsed with FCCP injection, the highest oxygen consumption occurred in complex IV of the ETC. At this time, the difference between maximal oxygen consumption and basal respiration was called spare respiratory capacity (SRC). The SRC is regarded as extra capacity available in cells to produce energy in response to increased stress [45]. The 28% increase in basal respiration compared to maximal respiration may indicate an impairment of mitochondrial energy metabolism in the prostate organoid due to LPS treatment (Figure 8). The increased basal respiration was stabilized to a normal level by the treatment of kaempferol. From this, it can be inferred that mitochondrial dysfunction recovery is achieved by kaempferol. It also increases mitochondrial function to maintain homeostasis and enhances the activity of antioxidant factors. An analysis of mitochondrial energy metabolism revealed that mitochondrial respiration and OCR results showed that kaempferol was effective in preventing mitochondrial damage. LPS treatment increased the OCR in prostate organoids; therefore, increased OCR raised ROS generation in ETC. The co-treatment of kaempferol reduced OCR and the subsequent reduction in ROS could be expected, which is led by improving mitochondrial function. In addition to restoring mitochondrial activity via Nrf2 revealed in the current study, kaempferol promotes the apoptosis of LNCaP cells in a dose-dependent manner in the presence of dihydrotestosterone (DHT) [46]. In BPH, kaempferol is known to suppress tumor development by inhibiting the activity of the androgen receptor (AR) and exerting a significant anti-benign effect [47].

The significance of this study is that it tested the activity of kaempferol, a phytochemical, using prostate organoids. Recently, interest in organoids as a test platform for new drugs or bioactive substances has increased, but a search of recent related literature shows that most of them are reviews or are about organoid fabrication and establishment, and there are only a few cases of application in actual experiments [23,24]. 

In the present study, we treated prostate organoids with LPS to induce ROS. As shown in Figure 6, the correlation between mitophagy and mitochondria biogenesis was observed in the tissue, which could not be obtained in the cell culture system. In summary, kaempferol exerted a protective effect against LPS-induced inflammatory factors and ROS in prostate organoids, providing new evidence to support the Nrf2-based enhancement of mitochondrial function and beneficial effects via the maintenance of homeostasis.

## Figures and Tables

**Figure 1 foods-12-03836-f001:**
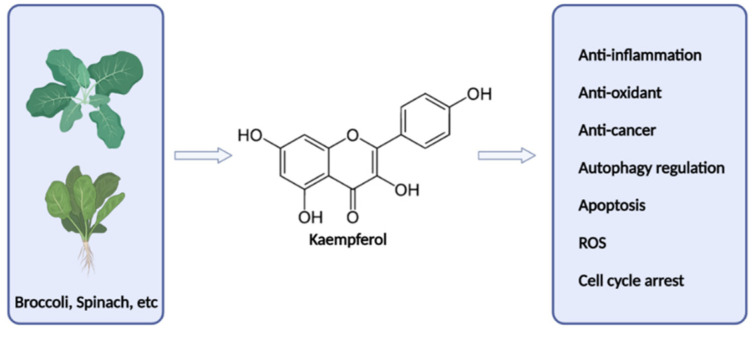
Kaempferol an overview.

**Figure 2 foods-12-03836-f002:**
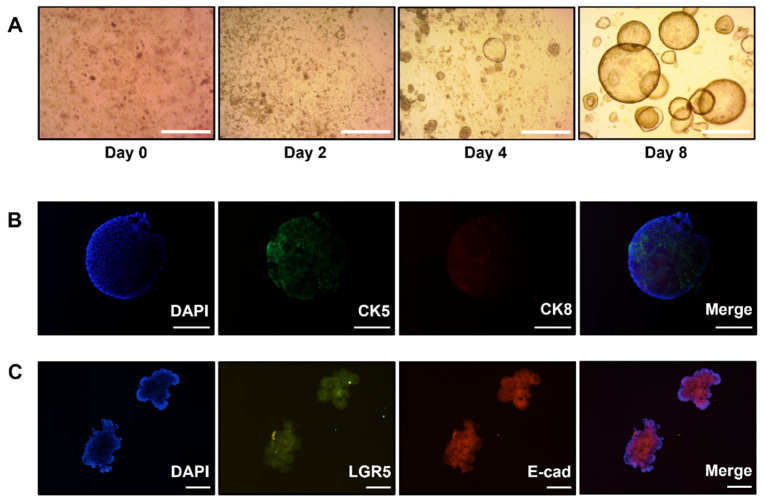
Establishment of mouse prostate organoid. Tissues were isolated from the prostate of a 7-week-old mouse and cultured in a Matrigel together with an organoid growth medium. Representative pictures of organoids 0, 2, 4, and 8 days after plating (**A**). The formed organoids were confirmed by monitoring under an Olympus CKX53 microscope. Scale bar, 300 μm. (**B**) Basal CK5 (green) and luminal CK8 (red) of organoids. Indigenous basal and luminal cell populations exist in organoids. (**C**) Expression of Lgr5 (green) and epithelial E-cadherin (red) in organoids. It was stained after 3–4 days of culture and observed with an Olympus BX53 fluorescence microscope, and nuclei were stained using DAPI. Scale bar, 300 μm. CK5, cytokeratin 5; CK8, cytokeratin 8; Lgr5, leucine-rich repeat-containing G-protein coupled receptor 5; DAPI, 4′,6-diamidino-2-phenylindole dihydrochloride.

**Figure 3 foods-12-03836-f003:**
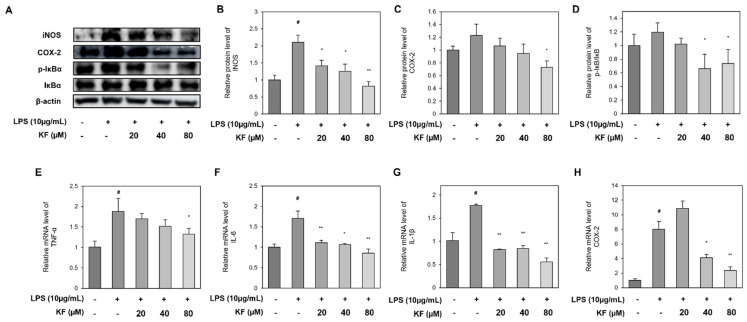
Kaempferol inhibits iNOS, COX-2, phosphorylation of IκB, and pro-inflammatory cytokines in LPS-induced prostate organoids. Western blotting results of (**A**) iNOS, COX-2, and p-IκBα/IκBα. Relative protein level of (**B**) iNOS, (**C**) COX-2, and (**D**) ratio of p-IκBα/IκBα. mRNA expression levels of (**E**) TNF-α, (**F**) IL-6, (**G**) IL-1β, and (**H**) COX-2. Each graph represents the relative value of the control group. Data represent mean ± SD from three independent experiments. # *p* < 0.05 vs. control group, * *p* < 0.05 and ** *p* < 0.01 vs. LPS-treated group. iNOS, inducible nitric oxide synthase; COX-2, cyclooxygenase 2; TNF-α, tubor necrosis factor-α; IL-6, interleukin-6; IL-1β, interleukin-1β; LPS, lipopolysaccharide.

**Figure 4 foods-12-03836-f004:**
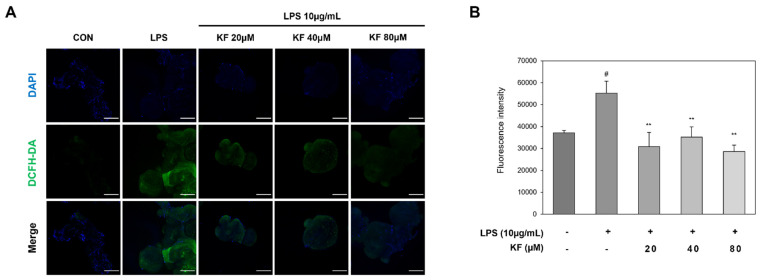
Intracellular ROS levels in prostate organoids were measured using H2DCFDA. Organoids were pretreated with kaempferol for 24 h, and then 10 μg/mL of LPS and 20, 40, and 80 μM of kaempferol were co-treated in each well for 24 h. (**A**) Intracellular ROS production was confirmed via DCF-DA fluorescence expression. (**B**) A graph representing the quantified data in Figure (**A**). Data were measured from three independent experiments and represent mean ± SD. # *p* < 0.05 vs. control group, # *p* < 0.05, and ** *p* < 0.01 vs. LPS-treated group. Scale bar, 300 μm. ROS, reactive oxygen species; DCF-DA, 2′,7′-dichlorodihydrofluorescein diacetate; LPS, lipopolysaccharide.

**Figure 5 foods-12-03836-f005:**
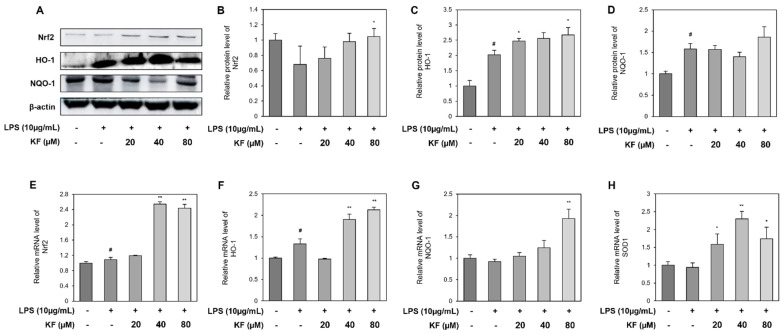
Kaempferol upregulates antioxidant responses by increasing Nrf2-related factor expressions in prostate organoids. Organoids were pretreated with kaempferol for 24 h, and then 10 μg/mL of LPS and 20, 40, and 80 μM of kaempferol were co-treated in each well for 24 h. Western blotting results of (**A**) Nrf2, HO-1, and NQO-1. Relative protein level of (**B**) Nrf2, (**C**) HO-1, and (**D**) NQO-1. mRNA expression levels of (**E**) Nrf2, (**F**) HO-1, (**G**) NQO-1, and (**H**) SOD1. Each graph represents the relative value of the control group. Data represent mean ± SD from three independent experiments. # *p* < 0.05 vs. control group, * *p* < 0.05, and ** *p* < 0.01 vs. LPS-treated group. Nrf2, nuclear erythrocyte 2-associated factor 2; HO-1, heme oxygenase-1; NQO-1, NAD(P)H quinone oxidoreductase 1; SOD1, superoxide dismutase1; LPS, lipopolysaccharide.

**Figure 6 foods-12-03836-f006:**
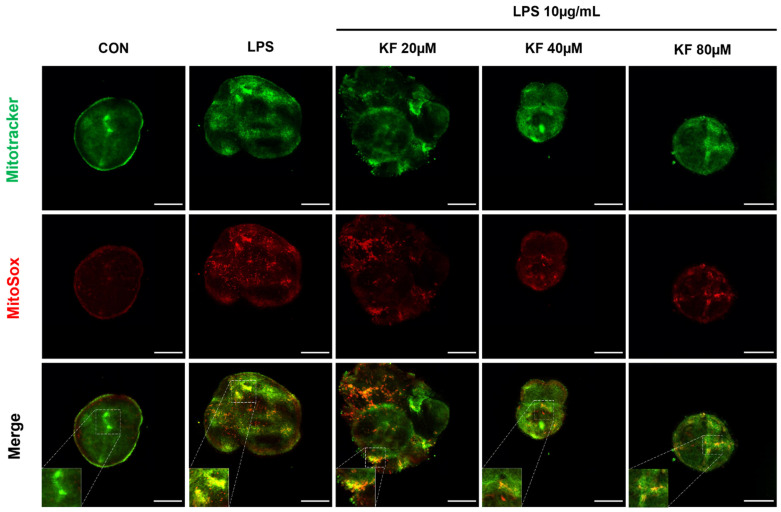
Kaempferol reduces mitochondrial ROS production in LPS-stimulated prostate organoids. Organoids were stained with MitoTracker Green and MitoSox Red, and the combined images indicate ROS generation in mitochondria. Representative images were obtained on a Nikon Eclipse C1si (20X). Scale bar, 300 μm. ROS, reactive oxygen species; LPS, lipopolysaccharide.

**Figure 7 foods-12-03836-f007:**
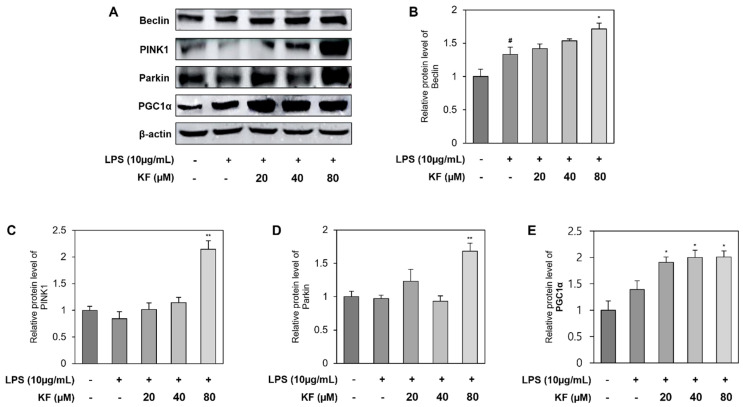
Kaempferol maintains homeostasis by improving mitochondrial function. Western blotting results of (**A**) Beclin, PINK1, Parkin, and PGC1α. Relative protein level of (**B**) Beclin, (**C**) PINK1, (**D**) Parkin, and (**E**) PGC1α. Each graph represents the relative value of the control group. Data represent mean ± SD from three independent experiments. # *p* < 0.05 vs. control group, * *p* < 0.05, and ** *p* < 0.01 vs. LPS-treated group. PINK1, PTEN-induced kinase 1; PGC1α, peroxisome proliferator-activated receptor gamma coactivator 1-alpha; LPS, lipopolysaccharide.

**Figure 8 foods-12-03836-f008:**
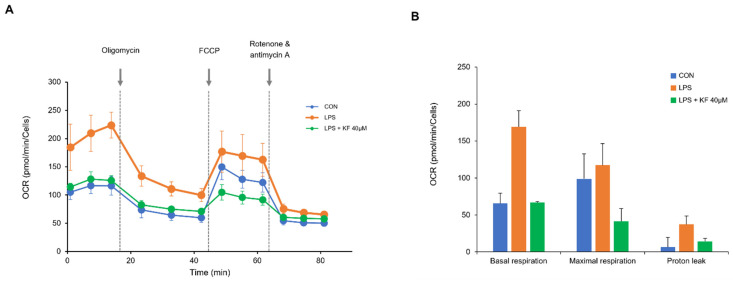
Effects of kaempferol and LPS on oxygen consumption. OCR was measured in real time using a metabolic analyzer and mitochondrial respiration inhibitors were sequentially injected. (**A**) Oxygen consumption rates of LPS-treated prostate organoids and kaempferol co-treatment. (**B**) Basal respiration, maximal respiration, and proton leak were measured simultaneously using OCR measurements. Data represent mean ± SD from four independent experiments. LPS, lipopolysaccharide; OCR, oxygen consumption rate.

**Table 1 foods-12-03836-t001:** Organoid medium composition.

Ingredient	Final Concentration
R-spondin conditioned medium	10%
Noggin conditioned medium	25%
N-acetyl-L-cysteine	1.25 mM
EGF	10 μM
DHT	50 ng/mL
A83-01	1 nM
HEPES	200 nM
Glutamax	10 mM
B27 Supplement	2 mM
Glutamax	2%
Y-27632	10 μM (only 5–7 days after passage)

**Table 2 foods-12-03836-t002:** Primer sequence for RT-qPCR.

Gene	Primer Sequence	
GAPDH	F	5′-AACAGCAACTCCCACTCTTC-3′
	R	5′-GTGGTCCAGGGTTTCTTACTC-3′
TNF-a	F	5′-AGCCCCCAGTCTGTATCCTT-3′
	R	5′-GAGGCAACCTGACCACTCTC-3′
IL-6	F	5′-GCCAGAGTCCTTCAGAGAGATA-3′
	R	5′-CAAACCTAGTGCGTTATGCCTA-3′
IL-1B	F	5′-CTGCTTCCAAACCTTTGACC-3′
	R	5′-AGCTTCTCCACAGCCACAAT-3′
COX-2	F	5′-CCAGATGCTATCTTTGGGGA-3′
	R	5′-GCTCGGCTTCCAGTATTGAG-3′
Nrf2	F	5′-TCCGCTGCCATCAGTCAGTC-3′
	R	5′-ATTGTGCCTTCAGCGTGCTTC-3′
HO-1	F	5′-AACAAGCAGAACCCAGTCTATGC-3′
	R	5′-AGGTAGCGGGTATATGCGTGGGCC-3′
NQO-1	F	5′-TTCTGTGGCTTCCAGGTCTT-3′
	R	5′-AGGCTGCTTGGAGCAAAATA-3′
SOD1	F	5′-TGGGTTCCACGTCCATCAGTA-3′
	R	5′-ACCGTCCTTTCCAGCAGTCA-3′

## Data Availability

The data used to support the findings of this study can be made available by the corresponding author upon request.

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
