# Peer review of "Kaempferol Alleviates Mitochondrial Damage by Reducing Mitochondrial Reactive Oxygen Species Production in Lipopolysaccharide-Induced Prostate Organoids"

_foods, 2023, doi:10.3390/foods12203836_

Round 1
Reviewer 1 Report
Dear Author, the submitted manuscript is relevant, however, it is suggested to make the following changes
It is suggested to include more information on the antioxidant capacity of Kaempferol in which it is indicated "thanks to its hydroxyl groups" Line 75.
It is suggested to deepen the described information on the imbalance generated by mitochondrial proliferation and degradation in the processes of apoptosis, ROS accumulation, and decreased mitochondrial DNA (Lines 82,83).
the methodology section
2. Organoid formation
Integrate the information of what type of growth factors were added or indicate if they are those found in LA Table 2 (Line 126).
Check in the different sections of methodology that the carbon dioxide formula is properly written, e.g. line 141, 214, 233, etc.
The results section
ROS increases prostate organoid oxygen consumption rate (OCR), and co-treatment of 411 kaempferol restores mitochondrial dysfunction through mitophagosomes.
It is not clear why only the 40 µM concentration of Kaempferol was used, it could be explained and included.
The conclusion section
I think the idea of including an image representing the conclusion of the paper may be useful visually, however, both the image and the text should conclusively describe the work done and I felt that both should be improved because they do not clearly describe the conclusion.
Author Response
Reviewer #1
- It is suggested to include more information on the antioxidant capacity of Kaempferol in which it is indicated "thanks to its hydroxyl groups" Line 75.
Response: Thank you for the reviewer’s suggestion. Based on your opinion, we added the comment in the manuscript, and revised content was highlighted in red so that editors and reviewers could easily identify changes (Line 74-75 in revised manuscript). Kaempferol, shown in [Figure 1], has four hydroxyl groups and is known to have antioxidant activity. However, in the structure of a flavonoid, it is important which sugar the hydroxyl group is located on.
- It is suggested to deepen the described information on the imbalance generated by mitochondrial proliferation and degradation in the processes of apoptosis, ROS accumulation, and decreased mitochondrial DNA (Lines 82,83).
Response: We apologize for our negligence in missing detail. Mitochondria are organs susceptible to oxidative damage, and imbalance caused by mitochondrial dysfunction can increase the amount of damaged mitochondria in the cell, reducing metabolic efficiency and generating excessive ROS, which ultimately leads to cell death. We found additional related references and added the comment in the manuscript based on the references, and revised content was highlighted in red too (Line 81-87 in revised manuscript).
- Integrate the information of what type of growth factors were added or indicate if they are those found in LA Table 2 (Line 126).
Response: This was a point where there could be confusion about badge composition. We have modified some text and added Y-27632 in [Table 1]. (Line 128-129, [Table 1])
- Check in the different sections of methodology that the carbon dioxide formula is properly written, e.g. line 141, 214, 233, etc.
Response: Thanks for finding out our mistake. We have modified carbon dioxide properly written (Line 145, 218, 237, 255)
- It is not clear why only the 40 µM concentration of Kaempferol was used, it could be explained and included.
Response: Thank you for the question that captures the core of the manuscript. Since kaempferol was known to be cytotoxic at higher concentrations, typically above 50 ?m, it is very important to determine the optimal treat concentration. As can be seen in the figure presented above, Kaempferol showed significant anti-inflammatory, antioxidant, and ROS scavenging activity even at 40 ?m (Line 424-428 in revised manuscript).
- I think the idea of including an image representing the conclusion of the paper may be useful visually, however, both the image and the text should conclusively describe the work done and I felt that both should be improved because they do not clearly describe the conclusion.
Response: I think your suggestion is correct. Thank you for your suggestion. Although our intention to organize research results was good, we concluded that it was better to remove the figure because recent web-based image searches have raised the possibility that our specific results could be mistaken for general mechanisms. Instead, we described our results more clearly and in detail (Line 557-569 in revised manuscript).
Reviewer 2 Report
The present work evaluates the effect of kaempferol on prostate cancer trough the use of organoids.The manuscript is well written; the authors explain in detail the methodology used and justify the reason for the parameters measured.
Although it is known that polyphenols are able to activate the Nrf2 pathway and modified different pro inflammatory interleukins, there are few studies that evaluate it in the context of prostate cancer.
However authors are encouraged to add to the discussion references to other studies that involucrate the use of polyphenols in prostate cancer and in others types of cancer.
There is a lack of discussion with recent bibliography.
Author Response
Reviewer #2
- The present work evaluates the effect of kaempferol on prostate cancer through the use of organoids.
The manuscript is well written; the authors explain in detail the methodology used and justify the reason for the parameters measured.
Although it is known that polyphenols are able to activate the Nrf2 pathway and modified different pro inflammatory interleukins, there are few studies that evaluate it in the context of prostate cancer.
However authors are encouraged to add to the discussion references to other studies that involucrate the use of polyphenols in prostate cancer and in others types of cancer.
There is a lack of discussion with recent bibliography.
Response: Thank you for the reviewer’s suggestion. Based on your opinion, we added the comment in the manuscript, and revised content was highlighted in red so that editors and reviewers could easily identify changes (Line 551-556 in revised manuscript). Kaempferol has been shown to inhibit cell proliferation in prostate cancer cells and promote apoptosis, particularly by inhibiting angiogenesis, and has shown anticancer activity in several other cancer cells. If the low bioavailability of kaempferol can be overcome, it is expected to play a role as a flavonoid therapeutic agent.
Reviewer 3 Report
This very interesting article shows how powerful organoids can be to study the effect of different phytochemicals on physiological processes. In this article, the protective role of kaempherol against oxidative stress is specifically studied. The experimental strategy used is correct, the results are clearly presented and the conclusions seem consistent with them.
Just a comment, I could be wrong but shouldn't the title be something like "Kaempferol Alleviates the LPS-induced Mitochondrial Damage by Reducing Mitochondrial ROS Production in Prostate Organoids"?
Author Response
Reviewer #3
This very interesting article shows how powerful organoids can be to study the effect of different phytochemicals on physiological processes. In this article, the protective role of kaempherol against oxidative stress is specifically studied. The experimental strategy used is correct, the results are clearly presented and the conclusions seem consistent with them.
- Just a comment, I could be wrong but shouldn't the title be something like "Kaempferol Alleviates the LPS-induced Mitochondrial Damage by Reducing Mitochondrial ROS Production in Prostate Organoids"?
Response: We really appreciate your precious suggestion. If the main focus is kaempherol, it would be correct to change the title as you suggested, however, we would like to emphasize the establishment of the "LPS-induced prostate organoid" model.
Therefore, if you don't mind, it would be better to keep the current title.
Round 2
Reviewer 2 Report
The authors have improved the manuscript according to the reviewers' suggestions.
Author Response
Thank you again for your suggestion to improve the scientific quality of our article.